# Offers of Cigarettes and E-Cigarettes Among High School Students: A Population Study from California

**DOI:** 10.3390/ijerph16071143

**Published:** 2019-03-30

**Authors:** Adam G. Cole, Sharon E. Cummins, Shu-Hong Zhu

**Affiliations:** 1Moores Cancer Center, University of California San Diego, San Diego, CA 92093, USA; acole@ucsd.edu; 2Department of Family Medicine and Public Health, University of California San Diego, San Diego, CA 92093, USA; scummins@ucsd.edu

**Keywords:** electronic cigarettes, tobacco control, youth tobacco use, risk for tobacco use, susceptibility to future smoking

## Abstract

Receiving offers of cigarettes or e-cigarettes can contribute to the progression from intention to actual use. However, there is a lack of data about the prevalence or characteristics of youth being offered cigarettes or e-cigarettes. A random sample of 91 high schools in the state of California (with 40,137 students) participated in the 2015–16 California Student Tobacco Survey. Offers of cigarettes and e-cigarettes in the last 30 days were assessed. Separate multilevel logistical regression models identified student characteristics associated with being offered cigarettes and e-cigarettes. On average, 11.1% and 16.1% of all students reported being offered cigarettes and e-cigarettes in the last 30 days, respectively. Among those who received offers of cigarettes, 45.5% were never smokers. Among those who received offers of e-cigarettes, 29.6% were never vapers. Male students were more likely to report being offered an e-cigarette than female students (Adjusted Odds Ratio (AOR) 1.13), as were students with friends that used e-cigarettes (AOR 5.14–23.31) and those with high sensation seeking tendencies (AOR 1.33). Similar characteristics were associated with offers of cigarettes. Including measures of offers of cigarettes and e-cigarettes in surveillance systems could help identify those at risk of future cigarette and e-cigarette use.

## 1. Introduction

Intentions are a strong predictor of behaviour [1], and measures that assess an adolescent’s intentions to smoke in the future are frequently used to identify those that are at risk of smoking. Pierce and colleagues [2,3] used a series of questions to identify and label adolescents who have never smoked cigarettes and are not committed to remaining smoke-free as susceptible to future smoking. The validity of this measure for identifying students at risk of cigarette and e-cigarette use has been supported in many studies [4,5,6,7]. However, even if the intention is present, access to these products is critical in order to try them. In the case of adolescents, access through retail channels is limited given age-restrictions for purchasing in many countries. Instead, many cigarette and e-cigarette users report getting tobacco products from peers [8,9,10] and an offer of cigarettes or e-cigarettes from others is likely a key step in the initiation process.

Longitudinal studies suggest that being offered cigarettes predicts later cigarette initiation and progression among youth [11,12]. Given this evidence, the receipt of offers of tobacco products within the last month may be a useful behavioral measure of risk of use in addition to cognitive measures of risk (i.e., susceptibility). However, to our knowledge no surveillance system currently measures the prevalence of offers of tobacco products, and only a single study has identified the frequency of cigarette offers to smoking and non-smoking youth [13]; evidence from this sample of adolescents in primary care in the USA indicate that approximately 1 in 4 non-current smokers and over 3 in 4 current smokers were offered a cigarette in the last month [13]. Data with respect to offers of e-cigarettes are currently lacking.

Given the gaps in the literature with respect to the prevalence of cigarette and e-cigarette offers and the characteristics of youth who receive offers for these products, the objectives of this study were to (1) measure the prevalence of youth being offered cigarettes and e-cigarettes in the last 30 days, and (2) identify the characteristics of those receiving product offers among a large, school-based, representative sample of students in California, USA.

## 2. Materials and Methods 

### 2.1. Design and Participants

This study used representative data collected as part of the 2015–16 cycle of the California Student Tobacco Survey (CSTS), a biennial survey that provides statewide estimates of tobacco use among middle school and high school students in California, USA. As described elsewhere [14], the CSTS uses a two-stage cluster sampling design in which school is the primary sampling unit and classroom is the secondary sampling unit. The state was divided into 12 regions, and the number of schools randomly selected within each region was proportional to the number of students in the region. The target population consisted of eighth, tenth, and twelfth grade students from public and non-sectarian schools; special education, juvenile court, district/county community, continuation, or other alternative schools were excluded. The current analyses focus on high school students (i.e., tenth and twelfth grades).

In 2015–16, the CSTS approached a random sample of 264 schools and 147 agreed to participate (55.7%). Among those that agreed, 119 schools completed the survey (81.0%) between October 2015 and June 2016 for a total of 47,981 participating students in eighth, tenth, and twelfth grades (mean student response rate: 75.5%). The current analyses included data from 91 high schools (27 middle schools were excluded, as was 1 high school with a response rate < 30%) that included 41,821 California students in tenth and twelfth grades. The CSTS survey was administered to students during class time either through a paper and pencil or online questionnaire available in English or Spanish; in 86.3% of schools, students completed the questionnaire online. A combination of active (i.e., signed permission forms) and passive (i.e., opt-out) permission protocols based on school district requirements were used to recruit students; the majority of school districts used passive permission protocols. Students also had the opportunity to decline participation on the day of the survey. The University of California San Diego Human Research Protection Program (#150256), the California State Committee for the Protection of Human Subjects (#15-04-6213), and appropriate school district committees approved all procedures.

### 2.2. Measures

The CSTS was designed to assess use of, knowledge of, and attitudes towards cigarettes and other tobacco products (e.g., e-cigarettes, hookah, cigarillos). The survey included questions that assessed use of different tobacco products, susceptibility to future use, social and environmental exposure to products, and known covariates of use. Offers of cigarettes and e-cigarettes (Yes/No) was assessed with a single question, “In the last 30 days, has anyone offered you…”, followed by a list of tobacco products, including cigarettes and e-cigarettes. Intention to use cigarettes and e-cigarettes in the future was assessed with a single question, “If one of your best friends offered you the following product, would you use it?”, followed by a list of tobacco products. Students could answer on a 4-point Likert scale, ranging from “definitely yes” to “definitely not”. Similar to other studies [4,5,6,7], *non-susceptible never users* responded “definitely not” to the question targeting cigarettes or e-cigarettes, while *susceptible never users* provided any other response to the question targeting cigarettes or e-cigarettes. Students were provided with a picture and description of the tobacco products in the survey. Cigarettes were “sold in packs and cartons. Popular brands include Marlboro, Newport, Pall Mall, Camel, and Winston.” E-cigarettes were “also called e-cigs, vapes, vape pens, e-hookah, hookah, pens, tanks or mods. Some come with liquid inside and others you fill yourself. Popular names are Blu, NJOY, MarkTen, eGo, Imperial, and Fantasia.” *Current users* reported using cigarettes or e-cigarettes on any days in the last 30, while *non-current users* indicated having used cigarettes or e-cigarettes but not in the last 30 days. Sensation seeking behavior was assessed with a single question—“I like new and exciting experiences, even if I have to break the rules”—on a 4-point Likert scale ranging from “strongly agree” to “strongly disagree”. Students indicating “strongly agree” or “agree” were grouped together, while those indicating “disagree” or “strongly disagree” were grouped together. The survey also collected demographic information (i.e., gender, grade, and ethnicity), the proportion of friends that smoke cigarettes and use e-cigarettes, and self-reported academic achievement in the last year.

### 2.3. Statistical Analyses

We used survey weights in the descriptive statistics and logistic regression models to produce population-based estimates throughout. Weights were calibrated to the state grade distribution so that the total of the survey weights by grade would be equal to the actual enrolments in each grade. Students with missing outcomes and/or predictors were excluded from the analysis (*n* = 1684 representing 4.0% of the total sample). The analysis first explored the prevalence of offers of cigarettes and the characteristics of students offered them in the last 30 days and then duplicated the analysis for offers of e-cigarettes. Descriptive analyses examined the characteristics of students that reported being offered cigarettes and e-cigarettes in the last 30 days while accounting for the design effect (i.e., regional strata and student-level clustering within schools). Two multilevel logistic regression models identified student-level demographic and behavioral characteristics associated with being offered cigarettes and e-cigarettes. All models accounted for student-level clustering within schools and included all other covariates. All analyses were performed using SAS software, Version 9.4 [15].

## 3. Results

### 3.1. Offers of Cigarettes

Weighted demographic characteristics of the sample are presented in Table 1. The gender and grade distribution were approximately equal and approximately half of students identified as Hispanic. An average of 11.1% of California students in tenth and twelfth grades reported being offered cigarettes. The weighted characteristics and correlates of students who reported being offered cigarettes in the last 30 days are presented in Table 2.

As shown in Table 2, susceptible never smokers, non-current smokers, and current smokers had higher odds of reporting offers of cigarettes in the last 30 days relative to non-susceptible never smokers (Adjusted Odds Ratio (AOR) 2.63, 5.08, and 47.68, respectively). Students with friends who smoke cigarettes also had higher odds of reporting offers of cigarettes in the last 30 days, relative to those with no friends that smoke cigarettes (AOR 4.02–10.53). Finally, students with high sensation seeking tendencies had higher odds of reporting offers of cigarettes in the last 30 days relative to those with low sensation seeking tendencies (AOR 1.64).

### 3.2. Offers of E-Cigarettes

An average of 16.1% of California students in tenth and twelfth grades reported being offered e-cigarettes in the last 30 days. The weighted characteristics and correlates of students who reported being offered cigarettes in the last 30 days are presented in Table 3. Similar to the results in Table 2, susceptible never users, non-current users, and current users had higher odds of reporting offers of e-cigarettes in the last 30 days relative to non-susceptible never users (AOR 2.97, 3.81, and 22.13, respectively). Students with friends that used e-cigarettes also had higher odds of reporting offers of e-cigarettes in the last 30 days, relative to those with no friends that used e-cigarettes (AOR 5.14–23.31). Finally, students with high sensation seeking tendencies had higher odds of reporting offers of e-cigarettes in the last 30 days relative to those with low sensation seeking tendencies (AOR 1.33).

### 3.3. Offers of Cigarettes and E-Cigarettes According to Susceptibility to Future Product Use

Although the measure of intention differentiates never-users into *susceptible* and *non-susceptible* categories based on their responses, there may be differences in offers according to specific responses since *susceptible never users* includes a range of intentions (i.e., “probably not”, “probably”, and “definitely”). As shown in Figure 1, a similar pattern is observed between being offered cigarettes and student responses to measures of susceptibility to future smoking, and between being offered e-cigarettes and student responses to measures of susceptibility to future e-cigarette use. A clear gradient is apparent, with a lower percentage of students reporting being offered each product who would “definitely not” use the product if their best friend offered it relative to students who would “probably” or “definitely” use the product if their best friend offered it. A lower percentage of students that reported ever using cigarettes and e-cigarettes reported receiving offers of the products (28.9% and 25.5%, respectively) compared to students who would “definitely yes” use the product if their best friend offered it (47.9% and 48.5%, respectively).

Although the rate of offers of cigarettes and e-cigarettes across groups is important to measure, it is also important to consider the proportion of offers to each group given they have different sample sizes. For example, most adolescents were never smokers. This means that even a small rate of cigarette offers in this subgroup would translate into a large number of students offered cigarettes. In fact, of all offers of cigarettes, 45.4% were given to never smokers, while 26.5% and 28.1% were to non-current and current cigarette smokers, respectively. With respect to all offers of e-cigarettes, 29.7% were given to never e-cigarette users, 36.0% to non-current e-cigarette users, and 34.4% to current e-cigarette users. 

## 4. Discussion

The results of the current study indicate that both users and non-users have been offered cigarettes and e-cigarettes, and the rates are substantial. More students reported receiving offers of e-cigarettes relative to cigarettes (16.1% and 11.1%, respectively). Although most current smokers reported receiving offers of cigarettes, some never users also reported receiving such offers. Furthermore, although a lower percentage of never smokers reported receiving product offers, the proportion of all offers to susceptible and non-susceptible never users was high since most students are never smokers. Similar patterns were identified for e-cigarette offers. The characteristics of students receiving offers of cigarettes and e-cigarettes were also quite similar. Including measures of offers of cigarettes and e-cigarettes in surveillance systems could not only provide an indication of the prevalence of offers of products among youth, but it could also aid in identifying those at risk of cigarette or e-cigarette use.

This study identified that more students reported receiving offers of e-cigarettes relative to cigarettes. Some of the difference in the prevalence of offers could be a result of the relative prevalence of each behaviour (i.e., cigarette smoking is much less prevalent than e-cigarette use [16]). Another explanation for this difference could result from variation in the perceived risk and social acceptability of each product among youth (e.g., e-cigarettes are perceived as safer and more socially acceptable than cigarettes [17,18]). Alternatively, there could be differences in the accessibility of products or situations where product offers occur (i.e., primarily social situations). Additional research is needed to identify where and when product offers occur and who is offering the product, which could improve prevention programs.

These data indicate that the prevalence of offers of cigarettes and e-cigarettes varied according to smoking and e-cigarette use status and according to susceptibility to future product use. Given that most cigarette and e-cigarette users report getting the products from peers [8,9,10], it is not surprising that many current smokers and e-cigarette users receive product offers. However, it is unclear why fewer non-current smokers and e-cigarette users receive product offers relative to those who have never used cigarettes or e-cigarettes but would “definitely” use the product if their best friend offered. It may be that offers of cigarettes or e-cigarettes are more memorable to never users relative to non-current users. Alternatively, it is possible that some non-current users may have tried smoking or using e-cigarettes, did not enjoy it, are clearly uninterested in using the products in the future, and have somehow communicated this to their peer group. Finally, there may be differences in the number of friends that use cigarettes and e-cigarettes between those most susceptible to future product use (i.e., respond “definitely yes”) and non-current users. Future research should explore potential reasons for the differences in product offers between these groups as there might be implications for how to increase the effectiveness of prevention programs.

Susceptible and even non-susceptible never users are at risk of future cigarette and e-cigarette use if they are receiving offers of these products. Longitudinal data indicate that non-smoking students who receive offers of cigarettes are more likely to initiate cigarette smoking [11,12]. However, differences in the risk of initiating smoking when offered cigarettes according to susceptibility to future smoking have not been explored in longitudinal studies. The current results suggest this may be an important investigation for future studies. A low percentage of students who would “definitely not” use cigarettes or e-cigarettes if their best friend offered it reported receiving offers relative to students who would “probably” or “definitely” use the product. Although these data are cross-sectional, it is possible that measures of offers of cigarettes and e-cigarettes may provide insight in identifying students at risk of initiating cigarette and e-cigarette use in addition to current measures of susceptibility. Longitudinal data are needed to evaluate the predictive ability of such measures to future product use and the discriminant validity compared to current measures of susceptibility to future product use. Furthermore, such data would help to clarify the potential bidirectional association that may exist between susceptibility to future product use and product offers.

The characteristics of students receiving offers of cigarettes and e-cigarettes were quite similar and resembled the characteristics of cigarette and e-cigarette users. For example, male and older students had higher odds of receiving offers of cigarettes and e-cigarettes. Not surprisingly, cigarette smoking or e-cigarette use status was strongly associated with being offered a product in the last 30 days, such that students that reported using cigarettes or e-cigarettes in the last 30 days had the highest odds of a product offer. Similarly, students with friends that smoke cigarettes or use e-cigarettes had higher odds of being offered each product. Research from Northern England indicates that offers of cigarettes are much more frequent among those whose best friend smokes [19]. Furthermore, being offered cigarettes, particularly by peers, could help maintain cigarette smoking; evidence indicates that not only are youth who receive offers of cigarettes more likely to transition to higher levels of use [12,20], but they are also less likely to quit smoking [11]. Longitudinal data for the association between offers of e-cigarettes and e-cigarette use progression and maintenance are currently lacking, representing an area to explore in future research. Students who reported sensation seeking attitudes were also more likely to report receiving offers of both cigarettes and e-cigarettes. Although it cannot be confirmed with these cross-sectional data, it is possible that youth high in sensation seeking attitudes join peer groups that engage in cigarette smoking or e-cigarette use in order to receive offers of the products. Future research should explore how youth access cigarettes and e-cigarettes through their social networks and the nature of offers being made to youth.

### Limitations and Strengths

The cross-sectional design prevents us from making any causal inferences about receiving offers of cigarettes or e-cigarettes and product initiation or progression. We are also not able to test the predictive validity or discriminant validity of this measure against other measures of intention to smoke (such as susceptibility to future smoking). The current measure only asked if students received any offers of cigarettes or e-cigarettes in the last 30 days and did not ask about the number of offers received within the last 30 days or who made the offers of products (e.g., friends, older peers, family members, or strangers). To our knowledge, this was the first study to identify the prevalence of offers of e-cigarettes and characteristics of students that receive offers of e-cigarettes. The low cigarette and e-cigarette prevalence in California relative to other jurisdictions may limit the generalizability of these results; therefore, data with respect to offers of tobacco products from other states and countries are needed. The present study included a large sample size that was representative of students in tenth and twelfth grades from across California, which is a significant strength. The CSTS is a state survey that occurs every two years, and the repeated nature of the survey will allow researchers to monitor changes in the prevalence of offers of cigarettes and e-cigarettes over time.

## 5. Conclusions

A substantial percentage of students, both users and non-users, reported being offered cigarettes, and an even greater percentage of them reported being offered e-cigarettes. Including measures of offers of cigarettes and e-cigarettes in usual surveillance tools would provide continued monitoring of this behavior. Students who reported using cigarettes or e-cigarettes, having friends that used cigarettes or e-cigarettes, and high sensation seeking attitudes had higher odds of receiving offers of cigarettes and e-cigarettes. Asking students about offers of cigarettes and e-cigarettes may be a useful behavioral measure to identify those at risk of future tobacco use in addition to current measures of cognitive susceptibility. 

## Figures and Tables

**Figure 1 ijerph-16-01143-f001:**
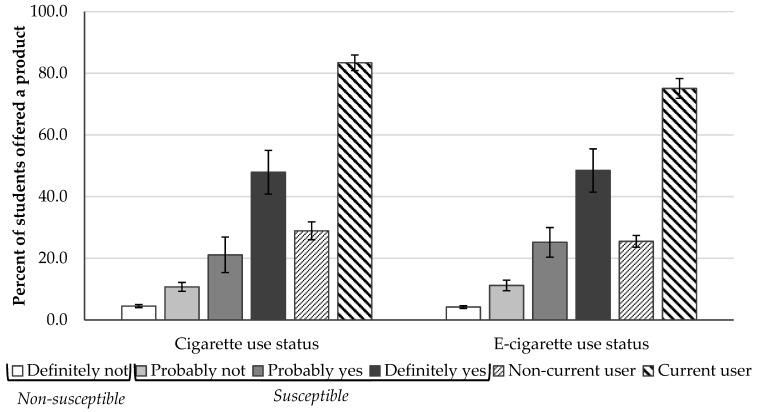
High school students who reported being offered cigarettes and e-cigarettes in the last 30 days according to their responses to measures of intentions to future use and current product use, 2015–16 California Student Tobacco Survey (*n* = 40,137).

**Table 1 ijerph-16-01143-t001:** Weighted demographic characteristics of the sample of high school students, 2015–16 California Student Tobacco Survey (*n* = 40,137).

Student Characteristic	% of Students (95% CI)
Gender	
Female	50.7 (49.2–52.1)
Male	49.3 (47.9–50.8)
Grade	
10	51.3 (48.4–54.2)
12	48.7 (45.8–51.6)
Ethnicity	
Non-Hispanic White	16.7 (13.2–20.2)
Non-Hispanic Black	3.4 (2.2–4.6)
Hispanic	54.9 (49.3–60.6)
Non-Hispanic Asian	13.4 (9.3–17.6)
Non-Hispanic American Indian or Alaska Native / Native Hawaiian or Other Pacific Islander / Other	3.3 (2.7–3.9)
Non-Hispanic Multiple	8.2 (7.2–9.2)
Cigarette smoking status	
Non-susceptible never smoker	74.1 (72.9-75.3)
Susceptible never smoker	12.0 (11.2–12.8)
Non-current smoker	10.2 (9.5–10.9)
Current smoker	3.7 (3.3–4.2)
E-cigarette use status	
Non-susceptible never user	55.0 (53.2–56.9)
Susceptible never user	14.8 (13.9–15.7)
Non-current user	22.8 (20.9–24.6)
Current user	7.4 (6.5–8.3)
Proportion of friends that smoke cigarettes	
None	55.8 (53.7–57.8)
Some	38.5 (36.9–40.1)
Most	4.2 (3.7–4.7)
All	1.5 (1.3–1.8)
Proportion of friends that use e-cigarettes	
None	52.1 (49.9–54.3)
Some	37.8 (36.6–39.1)
Most	8.0 (6.9–9.1)
All	2.0 (1.7–2.3)
Sensation seeking	
Disagree	48.3 (47.0–49.6)
Agree	51.7 (50.4–53.0)
Academic achievement	
Mostly As and Bs	52.9 (49.6–56.2)
Mostly Bs and Cs	32.8 (30.9–34.8)
Mostly Cs and Ds	10.7 (9.5–12.0)
Mostly Ds and Fs	3.5 (3.0–4.1)

**Table 2 ijerph-16-01143-t002:** Weighted characteristics and correlates of students who reported being offered cigarettes in the last 30 days, 2015–16 California Student Tobacco Survey (*n* = 40,137).

Student Characteristic	% of Students (95% CI)	Adjusted Odds Ratio (95% CI) ^a^
Overall	11.1 (10.1–12.2)	–
Gender		
Female	9.7 (8.5–10.9)	1.00
Male	12.6 (11.5–13.8)	1.21 (1.19, 1.23) *
Grade		
10	8.8 (7.7–9.9)	1.00
12	13.5 (12.3–14.8)	1.36 (1.34, 1.39) *
Ethnicity		
Non-Hispanic White	14.7 (13.4–16.1)	1.00
Non-Hispanic Black	7.0 (5.5–8.5)	0.71 (0.66, 0.75) *
Hispanic	11.4 (10.0–12.8)	0.81 (0.79, 0.84) *
Non-Hispanic Asian	5.6 (4.7–6.5)	0.53 (0.51, 0.55) *
Non-Hispanic American Indian or Alaska Native/Native Hawaiian or Other Pacific Islander/Other	13.0 (10.3–15.7)	0.84 (0.80, 0.88) *
Non-Hispanic Multiple	11.8 (10.0–13.6)	0.82 (0.79, 0.86) *
Cigarette smoking status		
Non-susceptible never smoker	4.5 (4.0–5.0)	1.00
Susceptible never smoker	14.4 (12.5–16.3)	2.63 (2.57, 2.69) *
Non-current smoker	28.9 (26.0–31.8)	5.08 (4.97, 5.20) *
Current smoker	83.4 (80.9–85.9)	47.68 (45.96, 49.46) *
Proportion of friends that smoke		
None	3.1 (2.7–3.5)	1.00
Some	18.1 (16.8–19.3)	4.02 (3.93, 4.11) *
Most	39.8 (34.7–44.9)	8.46 (8.18, 8.76) *
All	49.9 (44.1–55.7)	10.53 (10.03, 11.07) *
Sensation seeking		
Disagree	5.8 (5.0–6.6)	1.00
Agree	16.1 (14.7-17.6)	1.64 (1.61, 1.67) *
Academic achievement		
Mostly As and Bs	8.9 (8.0–9.8)	1.00
Mostly Bs and Cs	12.5 (11.0–14.0)	1.06 (1.04, 1.08) *
Mostly Cs and Ds	15.3 (13.4–17.2)	1.08 (1.05, 1.11) *
Mostly Ds and Fs	19.2 (16.6–21.8)	1.14 (1.09, 1.19) *

* Denotes significant differences (compared to the reference group) in the multilevel logistic regression model at *p* < 0.0001. ^a^ From a multilevel logistic regression model for being offered cigarettes in the past 30 days (*n* = 4580) versus not offered cigarettes in the last 30 days (*n* = 35,557), adjusting for all other covariates in the table.

**Table 3 ijerph-16-01143-t003:** Weighted characteristics and correlates of students who reported being offered e-cigarettes in the last 30 days, 2015–16 California Student Tobacco Survey (*n* = 40,137).

Student Characteristic	% of Students (95%CI)	Adjusted Odds Ratio (95% CI) ^a^
Overall	16.1 (14.7–17.5)	-
Gender		
Female	13.9 (12.4–15.3)	1.00
Male	18.4 (16.8–20.1)	1.13 (1.11, 1.15) *
Grade		
10	14.0 (12.5–15.6)	1.00
12	18.3 (16.7–19.9)	1.10 (1.08, 1.11) *
Ethnicity		
Non-Hispanic White	21.9 (18.6–25.2)	1.00
Non-Hispanic Black	10.5 (8.2–12.8)	0.72 (0.69, 0.76) *
Hispanic	15.9 (14.6–17.2)	0.79 (0.77, 0.81) *
Non-Hispanic Asian	9.8 (8.3–11.4)	0.64 (0.62, 0.66) *
Non-Hispanic American Indian or Alaska Native / Native Hawaiian or Other Pacific Islander / Other	15.8 (12.8–18.8)	0.73 (0.69, 0.76) *
Non-Hispanic Multiple	18.7 (16.7–20.7)	0.88 (0.85, 0.91) *
E-cigarette use status		
Non-susceptible never user	4.2 (3.7–4.6)	1.00
Susceptible never user	16.7 (14.6–18.8)	2.97 (2.90, 3.05) *
Non-current user	25.5 (23.6–27.4)	3.81 (3.73, 3.89) *
Current user	75.1 (71.9–78.3)	22.13 (21.53, 22.75) *
Proportion of friends that use e-cigarettes		
None	2.9 (2.6–3.2)	1.00
Some	22.6 (21.3–23.8)	5.14 (5.02, 5.25) *
Most	58.4 (55.1–61.6)	16.95 (16.49, 17.42) *
All	69.3 (65.0–73.7)	23.31 (22.28, 24.39) *
Sensation seeking		
No	9.3 (8.2–10.4)	1.00
Yes	22.5 (20.6–24.3)	1.33 (1.31, 1.35) *
Academic achievement		
Mostly As and Bs	13.8 (12.4–15.2)	1.00
Mostly Bs and Cs	17.7 (15.7–19.6)	1.01 (0.99, 1.02)
Mostly Cs and Ds	20.4 (18.3–22.6)	1.09 (1.07, 1.12) *
Mostly Ds and Fs	23.5 (20.8–26.2)	1.02 (0.98, 1.06)

* Denotes significant differences (compared to the reference group) in the multilevel logistic regression model at *p* < 0.0001. ^a^ From a multilevel logistic regression model for being offered e-cigarettes in the past 30 days (*n* = 6414) versus not offered e-cigarettes in the last 30 days (*n* = 33,723), adjusting for all other covariates in the table.

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
