# Peer review of "Offers of Cigarettes and E-Cigarettes Among High School Students: A Population Study from California"

_ijerph, 2019, doi:10.3390/ijerph16071143_

Round 1

Reviewer 1 Report

This cross-sectional analysis of the CSTS identifies the prevalence of cigarette and e-cigarette offers among high school students, as well as characteristics of those receiving offers. This is an interesting and well-written piece, which provides novel information about youth tobacco initiation. I have few minor comments below.

General

Did the survey include any items on actual/perceived access to tobacco products? If so, might be interesting to compare those data to offers.

Methods

Page 2, Line 72. Can you clarify whether parental permission was required for some or all schools? The line “A combination of permission protocols” is very vague; some example protocols would be useful for others conducting school-based surveys.

Page 2, Line 83. How was e-cigarettes defined in the survey? Were images or a definition provided?

Discussion

Page 6, Lines 174-178. One percentage is provided in this section, yet there are multiple areas where percentages would be helpful (e.g., line 174 after ‘common’; line 177 after ‘many’). Suggest adding other percentages or removing (83.4%).

Page 7, Lines 221-237. I was surprised by the academic achievement distribution across both cigarettes and e-cigarettes, particularly since anecdotally we are hearing that e-cigarette use is among more high achieving students. If space allows, perhaps a line or two discussing this finding, as well as potentially prioritizing these students for prevention.

Page 7, Line 238. The subheading “Limitations” might be more accurately written as “Limitations and Strengths.”

Conclusions

Page 8. Line 258. The resistance skills training conclusion, while important, and likely valid, does not seem drawn from the present manuscript based on introduction and framing. Perhaps consider something similar to what is written in the abstract (measurement).

Author Response

Response to Reviewer 1 comments:

Comment #1: General - Did the survey include any items on actual/perceived access to tobacco products? If so, might be interesting to compare those data to offers.

Our response: Thank you for your careful review of our manuscript. Unfortunately the survey did not include any items on actual/perceived access to tobacco products and as a result we are not able to explore the association in our paper. However, this would be an interesting avenue to explore in future research.

Comment #2: Methods - Page 2, Line 72. Can you clarify whether parental permission was required for some or all schools? The line “A combination of permission protocols” is very vague; some example protocols would be useful for others conducting school-based surveys.

Our response: Thank you for your comment. We have revised the manuscript to include additional information about the permission protocols used in the study and have referenced a more detailed technical report:

“A combination of active (i.e., signed permission forms) and passive (i.e., opt-out) permission protocols based on school district requirements were used to recruit students; the majority of school districts used passive permission protocols.”

Comment #3: Page 2, Line 83. How was e-cigarettes defined in the survey? Were images or a definition provided?

Our response: Thank you for your comment. Students were provided with a picture and description of the tobacco products (including e-cigarettes) included in the survey. The picture of e-cigarettes included mini (cigalike), mid-size (eGo), and mods. We have added the description use in the survey to the revised manuscript for clarity.

“Students were provided with a picture and description of the tobacco products included in the survey. Cigarettes were ‘sold in packs and cartons. Popular brands include Marlboro, Newport, Pall Mall, Camel, and Winston.’ E-cigarettes were ‘also called e-cigs, vapes, vape pens, e-hookah, hookah, pens, tanks or mods. Some come with liquid inside and others you fill yourself. Popular names are Blu, NJOY, MarkTen, eGo, Imperial, and Fantasia.’”

Comment #4: Discussion - Page 6, Lines 174-178. One percentage is provided in this section, yet there are multiple areas where percentages would be helpful (e.g., line 174 after ‘common’; line 177 after ‘many’). Suggest adding other percentages or removing (83.4%).

Our response: We have removed the percentage to be consistent throughout the discussion and for ease of reading.

Comment #5: Page 7, Lines 221-237. I was surprised by the academic achievement distribution across both cigarettes and e-cigarettes, particularly since anecdotally we are hearing that e-cigarette use is among more high achieving students. If space allows, perhaps a line or two discussing this finding, as well as potentially prioritizing these students for prevention.

Our response: Thank you for your comment. Although this is interesting, we prefer not to speculate about the findings, particularly since there is no consistent pattern for an association with offers of e-cigarettes.

Comment #7: Page 7, Line 238. The subheading “Limitations” might be more accurately written as “Limitations and Strengths.”

Our response: Thank you for the suggestion. We have updated the subheading.

Comment #8: Conclusions - Page 8. Line 258. The resistance skills training conclusion, while important, and likely valid, does not seem drawn from the present manuscript based on introduction and framing. Perhaps consider something similar to what is written in the abstract (measurement).

Our response: Thank you for your comment. We agree that this line in the conclusions is not drawn from the present data or discussed anywhere else in the manuscript, and we feel that it detracts from the overall findings. We have revised the conclusions to end with the following call-to-action:

“Including measures of offers of cigarettes and e-cigarettes in usual surveillance tools may help to identify those at risk of initiating cigarette and e-cigarette use.”

Reviewer 2 Report

This study is well written. I have a few comments that could improve the clarity of the manuscript. The discussion needs further work specifically regarding the significance of the findings – see comments below.

Abstract

Well written, I have no comments

Introduction

1.       For the first sentence – “Intentions are a strong predictor of behaviour” – please add a reference. You might want to link this also with a behaviour change theory such as the theory of planned behaviour which emphasizes intention as a precursor for changing behaviour.

Methods

2.       In your introduction you mention that Pierce has developed a valid measurement tool for susceptibility to cigarette smoking, and that this includes a series of questions. Yet you only use one question from this tool in your study to measure susceptibility. Could you explain this? This is a limitation of your study as the question you used may not be valid and truly reflect susceptibility.

Results

3.       I did not have access to supplementary table 1 so could not review this table.

4.       Table 1 and Table 2 could be combined into one table, but I leave this to the decision of the editor.

5.       Figure 1:

a.       There is a mistake in the differentiation of susceptible youth which should include the answers probably no, probably yes and definitely yes (and currently is pictured to include only definitely yes).

b.      I would delete the bars for current cig/e-cig use as they do not add anything to the data already provided in the tables.

c.       The x axis labelling is confusing – it should be susceptibility to cigarette smoking and susceptibility to e-cig use. The y-axis is the percentage offered cig/e-cig.

d.      The numbers of students in each susceptibility category is interesting data that right now is not provided – perhaps add this to figure 1 or on the text (apologies if this is available in supplemental table 1 – I could not view this at the time of the review)

6.       Results should only provide the results – comments on the results should be left to the discussion. So words like ‘Interestingly’ (line 154, page 5) should be removed.

7.       For this sentence – ‘Approximately 29.9%, 9.4%, 2.7%, and 3.4% of all offers of cigarettes were to students who would “definitely not”, “probably not”, “probably yes” and “definitely yes” smoke a cigarette if offered by one of their best friends, respectively’

This is opposite of the data in the table and figure that show that in susceptible youth a higher proportion report being offered a cig. Perhaps these are reversed?

Also I am not quite sure how these rates were obtained. If this is out of all those who were offered cigarettes – then these percentages should sum to 100% (because all offers of cig also had to answer the questions regarding susceptibility). This sentence and the similar one regarding e-cig are not clear. Same for the percentages regarding offers to current users or non-users. Participants could either be users or non-users so these percentages should sum to 100%.

If I understand correctly – these all sum together to 100%, so the first relate to percentages out of those that are never users. This should be made clear – “45.4%, 26.5% and 28.1% of all offers of cigarettes were given to students who were never users, non-current or current cigarette smokers, respectively.” ‘Approximately 29.9%, 9.4%, 2.7%, and 3.4% of all offers of cigarettes were to never users students who would “definitely not”, “probably not”, “probably yes” and “definitely yes” smoke a cigarette if offered by one of their best friends, respectively’.

Appropriate changes should be made also to the data regarding e-cig.

Discussion:

8.       You mention very briefly that your findings might have implications for increasing the effectiveness of prevention programs. This should be elaborated on – are current prevention programs tailored differently to susceptible youth? Has this been shown to be effective?

9.       You should add to the discussion on your findings that show an association between reporting receiving an offer of cig/e-cig and certain socio-demographic variables – male, older, white participants,  with higher rates of friends who use these products, who are sensation seekers and have a lower academic achievement, reported being offered cig/e-cig at a higher rate. Except for the ethnicity, these closely resemble the socio-demographic distribution of who smokes… supporting the notion that those who get offered are at a higher risk of becoming smokers themselves. The higher rate of offering white participants is surprising, no? Smoking rates are usually higher among minority groups.

10.   Under the limitations section – as far as I am aware, California is the leading state in the USA on tobacco control measures and has the lowest smoking rates in the US. This has implications on the generalizability of this data to other youth in the US or internationally.

11.   The first sentence under the limitations section – “There are limitations with the present study” can be removed. All studies have limitations and it is clear from the section heading that you are discussion limitations.

Conclusion

12.   Your last sentence is the first time you mention this – “cessation programs should emphasize resistance skills training given that many students receive offers of cigarettes or e-cigarettes.” This should be elaborated on in your discussion (see comment #8).

13.   Also you provide no data regarding whether cessation programs with resistance skills are effective to prevent youth smoking when offered cig/e-cig. Therefore this sentence should be rewritten to be more speculative such as programs that include resistance skills might be effective…

14.   Your conclusion lacks a main finding of your study which is included in your abstract – “Including measures of offers of cigarettes and e-cigarettes in surveillance systems could help identify those at risk of future cigarette and e-cigarette use.”

Author Response

Response to Reviewer 2 comments:

Comment #1: Introduction - For the first sentence – “Intentions are a strong predictor of behaviour” – please add a reference. You might want to link this also with a behaviour change theory such as the theory of planned behaviour which emphasizes intention as a precursor for changing behaviour.

Our response: Thank you for your careful review of our manuscript and for the suggestion. We have included a reference to the Theory of Planned Behavior (Ajzen, 1991) following this statement.

Comment #2: Methods - In your introduction you mention that Pierce has developed a valid measurement tool for susceptibility to cigarette smoking, and that this includes a series of questions. Yet you only use one question from this tool in your study to measure susceptibility. Could you explain this? This is a limitation of your study as the question you used may not be valid and truly reflect susceptibility.

Our response: Thank you for the comment. The purpose of this manuscript is not to compare the measure of susceptibility to the measure of offers, particularly given these are cross-sectional data. Rather, our interest was in identifying the prevalence of offers of cigarettes and e-cigarettes, particularly among non-users of each product. We present the prevalence of offers according to responses to our measure of susceptibility to future smoking because it is interesting to note that the prevalence of offers follows a similar pattern – few students at lowest risk are offered cigarettes or e-cigarettes, while a higher percentage at higher risk are offered cigarettes and e-cigarettes. Other school-based surveys have used variations (between 1 and 3 questions) of Pierce’s measure of susceptibility.

Comment #3: Results - I did not have access to supplementary table 1 so could not review this table.

Our response: We apologize for this oversight. Supplementary Table 1, which presents demographics characteristics of the sample, has been added to the main manuscript following a request from another reviewer.

Comment #4: Results - Table 1 and Table 2 could be combined into one table, but I leave this to the decision of the editor.

Our response: Thank you for your comment. Our preference would be to keep the tables separate for ease in reading and interpreting the results. Furthermore, there is overlap in the sample (e.g., some students offered e-cigarettes may also report being offered cigarettes) and it would be inappropriate to directly compare these groups.

Comment #5a: Figure 1 - There is a mistake in the differentiation of susceptible youth which should include the answers probably no, probably yes and definitely yes (and currently is pictured to include only definitely yes).

Our response: We apologize for this oversight. The figure has been updated in the revised manuscript.

Comment #5b: Figure 1 - I would delete the bars for current cig/e-cig use as they do not add anything to the data already provided in the tables.

Our response: Thank you for your comment. Although we agree that these data are presented in the tables, we believe that they help to provide a clear picture of the trend in the prevalence of offers of cigarettes and e-cigarettes according to product use status.

Comment #5c: Figure 1 - The x axis labelling is confusing – it should be susceptibility to cigarette smoking and susceptibility to e-cig use. The y-axis is the percentage offered cig/e-cig.

Our response: Thank you for your comment and suggestion. We have updated the figure to include a y-axis label and changed the label of the x-axis.

Comment #5d: Figure 1 - The numbers of students in each susceptibility category is interesting data that right now is not provided – perhaps add this to figure 1 or on the text (apologies if this is available in supplemental table 1 – I could not view this at the time of the review).

Our response: Thank you for your suggestion. These data are currently summarized in the Results (Section 3.3, second paragraph). We have attempted to simplify and clarify these findings for readers based on feedback from multiple reviewers. We have also included the Supplementary Table in the main manuscript.

Comment #6: Results - Results should only provide the results – comments on the results should be left to the discussion. So words like ‘Interestingly’ (line 154, page 5) should be removed.

Our response: We have eliminated these words from the Results of the revised manuscript.

Comment #7: Results - For this sentence – ‘Approximately 29.9%, 9.4%, 2.7%, and 3.4% of all offers of cigarettes were to students who would “definitely not”, “probably not”, “probably yes” and “definitely yes” smoke a cigarette if offered by one of their best friends, respectively’

This is opposite of the data in the table and figure that show that in susceptible youth a higher proportion report being offered a cig. Perhaps these are reversed?

Also I am not quite sure how these rates were obtained. If this is out of all those who were offered cigarettes – then these percentages should sum to 100% (because all offers of cig also had to answer the questions regarding susceptibility). This sentence and the similar one regarding e-cig are not clear. Same for the percentages regarding offers to current users or non-users. Participants could either be users or non-users so these percentages should sum to 100%.

If I understand correctly – these all sum together to 100%, so the first relate to percentages out of those that are never users. This should be made clear – “45.4%, 26.5% and 28.1% of all offers of cigarettes were given to students who were never users, non-current or current cigarette smokers, respectively.” ‘Approximately 29.9%, 9.4%, 2.7%, and 3.4% of all offers of cigarettes were to never users students who would “definitely not”, “probably not”, “probably yes” and “definitely yes” smoke a cigarette if offered by one of their best friends, respectively’.

Appropriate changes should be made also to the data regarding e-cig.

Our response: Thank you for your comment and suggestion. We have attempted to simplify and clarify these important findings for readers based on feedback from multiple reviewers. We also believe that this revision clearly communicates our key finding that many offers of cigarettes and e-cigarettes are given to non-users:

Although the rate of offers of cigarettes and e-cigarettes across groups is important to measure, it is also important to consider the proportion of offers to each group given they have different sample sizes. For example, most adolescents were never smokers. This means that even a small rate of cigarette offers in this subgroup would translate into a large number of students offered cigarettes. In fact, of all offers of cigarettes, 45.4% were given to never smokers, while 26.5% and 28.1% were to non-current and current cigarette smokers, respectively. With respect to all offers of e-cigarettes, 29.7% were given to never e-cigarette users, 36.0% to non-current e-cigarette users, and 34.4% to current e-cigarette users.”

Comment #8: Discussion - You mention very briefly that your findings might have implications for increasing the effectiveness of prevention programs. This should be elaborated on – are current prevention programs tailored differently to susceptible youth? Has this been shown to be effective?

Our response: Thank you for your comment. Given the relative lack of data for tailored prevention programs and the fact that this is not a key message of our study, we have removed this statement from the revised manuscript and instead have focused on the need for and potential usefulness of measures of offers in current surveillance systems.

Comment #9: Discussion - You should add to the discussion on your findings that show an association between reporting receiving an offer of cig/e-cig and certain socio-demographic variables – male, older, white participants,  with higher rates of friends who use these products, who are sensation seekers and have a lower academic achievement, reported being offered cig/e-cig at a higher rate. Except for the ethnicity, these closely resemble the socio-demographic distribution of who smokes… supporting the notion that those who get offered are at a higher risk of becoming smokers themselves. The higher rate of offering white participants is surprising, no? Smoking rates are usually higher among minority groups.

Our response: Thank you for your suggestion. We have included some text in the revised manuscript that comments on the fact that the characteristics of students receiving offers resemble the characteristics of users. We have also noted that male and older students had higher odds of receiving offers of cigarettes and e-cigarettes in addition to the other significant predictors noted previously:

“The characteristics of students receiving offers of cigarettes and e-cigarettes were quite similar and resemble the characteristics of cigarette and e-cigarette users. For example, male and older students had higher odds of receiving offers of cigarettes and e-cigarettes.”

Comment #10: Discussion - Under the limitations section – as far as I am aware, California is the leading state in the USA on tobacco control measures and has the lowest smoking rates in the US. This has implications on the generalizability of this data to other youth in the US or internationally.

Our response: Thank you for your suggestion. We have added this limitation and noted that data from other jurisdictions are needed:

“The low cigarette and e-cigarette prevalence in California relative to other jurisdictions may limit the generalizability of these results; therefore, data with respect to offers of tobacco products from other states and countries are needed.”

Comment #11: Discussion - The first sentence under the limitations section – “There are limitations with the present study” can be removed. All studies have limitations and it is clear from the section heading that you are discussion limitations.

Our response: Thank you for your suggestion. We have removed this sentence from the revised manuscript.

Comment #12: Conclusion - Your last sentence is the first time you mention this – “cessation programs should emphasize resistance skills training given that many students receive offers of cigarettes or e-cigarettes.” This should be elaborated on in your discussion (see comment #8).

Our response: Thank you for your comment. Given the changes to the revised manuscript, we have removed this sentence from the conclusion and instead recommended including measures of offers of cigarettes and e-cigarettes in other surveillance systems:

A substantial percentage of students, both users and non-users, reported being offered cigarettes and e-cigarettes, and an even greater percentage of them reported being offered e-cigarettes. Including measures of offers of cigarettes and e-cigarettes in usual surveillance tools would provide continued monitoring of this behavior. Students who reported using cigarettes or e-cigarettes, having friends that used cigarettes or e-cigarettes, and high sensation seeking attitudes had higher odds of receiving offers of cigarettes and e-cigarettes. Asking students about offers of cigarettes and e-cigarettes may be a useful behavioral measure to identify those at risk of future tobacco use in addition to current measures of cognitive susceptibility.

Comment #13: Conclusion - Also you provide no data regarding whether cessation programs with resistance skills are effective to prevent youth smoking when offered cig/e-cig. Therefore this sentence should be rewritten to be more speculative such as programs that include resistance skills might be effective…

Our response: Thank you for your comment. We have removed references to resistance skills programs from the manuscript and have instead focused on recommending including measures of offers of tobacco products in other surveillance systems.

Comment #14: Conclusion - Your conclusion lacks a main finding of your study which is included in your abstract – “Including measures of offers of cigarettes and e-cigarettes in surveillance systems could help identify those at risk of future cigarette and e-cigarette use.”

Our response: Thank you for your comment. We have updated the conclusion to match this statement in the abstract and more closely align with the main findings of the manuscript.

Reviewer 3 Report

E-cigarettes have received much attention from researchers nowadays, and most studies have focused on e-cigarette product use among youth. This paper attempts to close a gap by exploring the behavior of offering the products to peers and friends by susceptibility and smoking status among high school students in California. The authors applied appropriate statistical models to a state representative dataset, and the paper is well written, providing clear results and informative discussion. A few suggestions might be considered by authors to further improve the manuscript.

1. Table 1 and 2: Please include 95% confidence intervals for the estimates of percentages of high-school students who reported being offered cigarettes and e-cigarettes by groups, as well as statistical significance indicated by p values.

2. Readers are probably curious about student reaction when being offered cigarettes and e-cigarettes, including “definitely not”, “probably not”, “probably yes”, “definitely yes”, among never user, non-current users, and current users, respectively. Those percentages would help to contextualize the results better. The differences between never user and non-current users may have implications on prevention of re-initiation of tobacco and e-cigarette use.

3. Please add error bars in Figure 1 so that readers would have a sense of whether this association is statistically significant or not.

4. One recent study has used a nationally representative sample PATH to examine e-cigarette offers from friends among youth. Please cite the article and reframe your contribution.

Sawdey, M. D., Day, H. R., Coleman, B., Gardner, L. D., Johnson, S. E., Limpert, J., ... & Pearson, J. L. (2019). Associations of risk factors of e-cigarette and cigarette use and susceptibility to use among baseline PATH study youth participants (2013–2014). Addictive behaviors, 91, 51-60.

Author Response

Response to Reviewer 3 comments:

Comment #1: Table 1 and 2: Please include 95% confidence intervals for the estimates of percentages of high-school students who reported being offered cigarettes and e-cigarettes by groups, as well as statistical significance indicated by p values.

Our response: Thank you for your careful review of our manuscript. We have updated the tables to include 95% confidence intervals for the prevalence estimates.

Comment #2: Readers are probably curious about student reaction when being offered cigarettes and e-cigarettes, including “definitely not”, “probably not”, “probably yes”, “definitely yes”, among never user, non-current users, and current users, respectively. Those percentages would help to contextualize the results better. The differences between never user and non-current users may have implications on prevention of re-initiation of tobacco and e-cigarette use.

Our response: Thank you for your comment. Unfortunately we are unable to comment on this.

Comment #3: Please add error bars in Figure 1 so that readers would have a sense of whether this association is statistically significant or not.

Our response: Thank you for your suggestion. We have updated the figure to include 95% confidence intervals for the prevalence estimates.

Comment #4: One recent study has used a nationally representative sample PATH to examine e-cigarette offers from friends among youth. Please cite the article and reframe your contribution.

Sawdey, M. D., Day, H. R., Coleman, B., Gardner, L. D., Johnson, S. E., Limpert, J., ... & Pearson, J. L. (2019). Associations of risk factors of e-cigarette and cigarette use and susceptibility to use among baseline PATH study youth participants (2013–2014). Addictive behaviors, 91, 51-60.

Our response: Thank you for the suggestion. Upon careful review of this article, it does not appear to provide any national data about the prevalence or correlates of offers of cigarettes and e-cigarettes.

Reviewer 4 Report

 This article describes rates of offers of cigarettes and e-cigarettes among California youth, as well as characteristics of students receiving offers. On the whole, I found the article to be well-written and believe it offers a contribution to the field.

Specific comments:

1. There is existing literature on sources of e-cigarettes, including offers from other people and sharing with other people (Kong et al., 2017; Pepper et al., 2018; and Meyers et al., 2017). Full citations appear at the end of these comments. The authors should include this as background and discuss their findings in the context of this prior research.

2. It would be helpful to understand whether the students who are being offered cigarettes are the same students being offered e-cigarettes. Please include a cross-tab and discuss the findings.

3. Readers may wish to better understand the multilevel model. What was the ICC and 95% confidence interval? How would you characterize the degree of clustering?

4. Please include Supplemental Table 1 (the respondent characteristics) as part of the main manuscript. If there is not room for an additional table in the manuscript, please report some key statistics from that table in the text. Most importantly, please report the percentages for each level of cigarette smoking status and e-cigarette use status in the body of the manuscript (regardless of whether Table 1 is included in the article or supplemental).

5. As far as I can tell, the second paragraph of section 3.3 is the same data from the first paragraph but with the direction flipped. That is, the first paragraph describes the % of students in different levels of susceptibility who received offers and the second paragraph is the % of students who receive offers by level of susceptibility. Is that the case? If so, I don’t think it is appropriate to report the same finding two different ways. I would suggest picking one or the other to present in the Results section. Alternative interpretations could be mentioned in the Discussion. If I am not correct in my interpretation of the results in section 3.3, the findings need to be described in a clearer manner.

6. Is there an error in Figure 1? I thought that the category “Definitely not” comprised non-susceptible and the other categories (probably not, probably yes, and definitely yes) comprise susceptibility, but this is not how the figure is labeled. Also, please label the Y axis and adjust the color scheme slightly: the 2 darkest bars are hard to distinguish when printed in B&W.

7. In multiple places, the authors describe offers of cigarettes and e-cigarettes as “common.” I would not describe 11% or 16% as “common.” Please rephrase.

 8. Please provide citations to back up the point that prevention and cessation programs should emphasize resistance skills training. What is the evidence base for recommending that approach? The only thing that comes to mind is the D.A.R.E. program, which was not successful.

9. I suggest the authors include a clearer discussion of the potential bidirectionality of the relationship between susceptibility and offers. Youth might receive offers because their peers recognize their susceptibility or youth might be susceptible because they receive offers.

10. A quick online search for the CSTS instrument shows that the survey includes items on perceived harm (both absolute and relative to cigarettes) and addictiveness of e-cigarettes. How doe these attitudes relate to susceptibility and offers? If we understand the attitudes of susceptible youth or youth who receive offers, that could suggest methods of intervention (e.g., campaigns that target those attitudes).

11. The section of the abstract regarding offers of “a product” is difficult to follow. I had to look to the tables to understand which product was being referred to and why there were ranges of ORs presented. Please present these findings in a clearer manner.

12. Sensation seeking is typically assessed as a scale, rather than a single item. Is there precedent for using a single item? Has that approach between validated? Also, “ease of interpretation” is not an appropriate reason to recode responses.

13. When proposing future directions for research, I suggest considering research that would better characterize who is doing the offering: friends? Family? Peers from their own social group or peers from other social groups? I believe that some of the articles on e-cigarettes discuss this issue. I’m not sure if the literature on offers of cigarettes discuss this issue.  

Citations:

Kong G, Morean ME, Cavallo DA, Camenga DR, Krishnan-Sarin S. Sources of electronic cigarette acquisition among adolescents in Connecticut. Tob Regul Sci. 2017;3(1):10-16.

Meyers MJ, Delucchi K, Halpern-Felsher B. Access to tobacco among California high school students: the role of family members, peers, and retail venues. J Adolesc Health. 2017;61(3): 385-388.

Pepper JK, Coats EM, Nonnemaker JM, Loomis BR. How Do Adolescents Get Their E-Cigarettes and Other Electronic Vaping Devices? Am J Health Promot. 2018 Aug 1:890117118790366. doi: 10.1177/0890117118790366.

Author Response

Response to Reviewer 4 comments:

Comment #1: There is existing literature on sources of e-cigarettes, including offers from other people and sharing with other people (Kong et al., 2017; Pepper et al., 2018; and Meyers et al., 2017). Full citations appear at the end of these comments. The authors should include this as background and discuss their findings in the context of this prior research.

Our response: Thank you for your careful review of our manuscript and for including suggested articles to strengthen the manuscript. We have briefly referenced the importance of social sources of cigarettes and e-cigarettes in both the introduction and discussion of the revised manuscript:

“In the case of adolescents, access through retail channels is limited given age-restrictions for purchasing in many countries. Instead, many cigarette and e-cigarette users report getting tobacco products from peers [8–10] and an offer of cigarettes or e-cigarettes from others is likely a key step in the initiation process.”

“These data indicate that the prevalence of offers of cigarettes and e-cigarettes varied according to smoking and e-cigarette use status and according to susceptibility to future product use. Given that most cigarette and e-cigarette users report getting the products from peers [8–10], it is not surprising that many current smokers and e-cigarette users receive product offers. However, it is unclear why fewer non-current smokers and e-cigarette users receive product offers relative to those who have never used cigarettes or e-cigarettes but would “definitely” use the product if their best friend offered.”

Comment #2: It would be helpful to understand whether the students who are being offered cigarettes are the same students being offered e-cigarettes. Please include a cross-tab and discuss the findings.

Our response: Thank you for your comment. Although we agree this investigation is important, we believe it is outside of the scope of the current manuscript. Given the much higher prevalence of e-cigarette use relative to cigarette use in California, it is likely that relatively more students who are offered cigarettes are also offered e-cigarettes than students who are offered e-cigarettes and are also offered cigarettes.

Comment #3: Readers may wish to better understand the multilevel model. What was the ICC and 95% confidence interval? How would you characterize the degree of clustering?

Our response: Thank you for your comment. We used multilevel modelling to account for the clustered nature of the data (i.e., students nested within schools) and not to specifically identify the amount of variation due to school-level factors. This type of modelling is common for studies using school-based samples. We believe including the ICC in the text is outside of the scope of the current paper as we were only interested in identifying student-level characteristics associated with being offered cigarettes and e-cigarettes. Other studies have calculated the ICC and identified significant between-school variability in cigarette smoking.

Comment #4: Please include Supplemental Table 1 (the respondent characteristics) as part of the main manuscript. If there is not room for an additional table in the manuscript, please report some key statistics from that table in the text. Most importantly, please report the percentages for each level of cigarette smoking status and e-cigarette use status in the body of the manuscript (regardless of whether Table 1 is included in the article or supplemental).

Our response: Thank you for your suggestion. We have included the Supplementary Table in the revised manuscript.

Comment #5: As far as I can tell, the second paragraph of section 3.3 is the same data from the first paragraph but with the direction flipped. That is, the first paragraph describes the % of students in different levels of susceptibility who received offers and the second paragraph is the % of students who receive offers by level of susceptibility. Is that the case? If so, I don’t think it is appropriate to report the same finding two different ways. I would suggest picking one or the other to present in the Results section. Alternative interpretations could be mentioned in the Discussion. If I am not correct in my interpretation of the results in section 3.3, the findings need to be described in a clearer manner.

Our response: Thank you for your comment. We believe it is important to highlight both the percentage of students that receive offers according to responses to susceptibility, and the proportion of offers to never users, non-current users, and current users since they present percentages based on different conditions of probability. We believe that both paragraphs in Section 3.3 provide an indication of the population-level impact. Based on feedback from other reviewers, we have simplified the results presented in the second paragraph:

Although the rate of offers of cigarettes and e-cigarettes across groups is important to measure, it is also important to consider the proportion of offers to each group given they have different sample sizes. For example, most adolescents were never smokers. This means that even a small rate of cigarette offers in this subgroup would translate into a large number of students offered cigarettes. In fact, of all offers of cigarettes, 45.4% were given to never smokers, while 26.5% and 28.1% were to non-current and current cigarette smokers, respectively. With respect to all offers of e-cigarettes, 29.7% were given to never e-cigarette users, 36.0% to non-current e-cigarette users, and 34.4% to current e-cigarette users.

Comment #6: Is there an error in Figure 1? I thought that the category “Definitely not” comprised non-susceptible and the other categories (probably not, probably yes, and definitely yes) comprise susceptibility, but this is not how the figure is labeled. Also, please label the Y axis and adjust the color scheme slightly: the 2 darkest bars are hard to distinguish when printed in B&W.

Our response: We apologize for this oversight. The figure has been updated in the revised manuscript. We have also labelled the y-axis and adjusted the fill of the bars to make them easier to distinguish from each other.

Comment #7: In multiple places, the authors describe offers of cigarettes and e-cigarettes as “common.” I would not describe 11% or 16% as “common.” Please rephrase.

Our response: Thank you for your comment. We have removed this phrasing from the revised manuscript.

Comment #8: Please provide citations to back up the point that prevention and cessation programs should emphasize resistance skills training. What is the evidence base for recommending that approach? The only thing that comes to mind is the D.A.R.E. program, which was not successful.

Our response: Thank you for your comment. Based on comments from other reviewers, we have removed the reference to resistance skills training in the manuscript. Instead, we have emphasized the potential usefulness of including measures of offers in usual tobacco surveillance systems.

Comment #9: I suggest the authors include a clearer discussion of the potential bidirectionality of the relationship between susceptibility and offers. Youth might receive offers because their peers recognize their susceptibility or youth might be susceptible because they receive offers.

Our response: Thank you for your suggestion. We have noted the cross-sectional nature of the data in the limitations and indicated that longitudinal studies are necessary to evaluate the predictive ability of measures of offers of cigarettes and e-cigarettes for identifying students at risk of future product use. We have added a sentence to the revised manuscript noting that longitudinal data are needed to clarify a potential bidirectional association:

“Longitudinal data are needed to evaluate the predictive ability of such measures to future product use and the discriminant validity compared to current measures of susceptibility to future product use. Furthermore, such data would help to clarify the potential bidirectional association that may exist between susceptibility to future product use and product offers.”

Comment #10: A quick online search for the CSTS instrument shows that the survey includes items on perceived harm (both absolute and relative to cigarettes) and addictiveness of e-cigarettes. How doe these attitudes relate to susceptibility and offers? If we understand the attitudes of susceptible youth or youth who receive offers, that could suggest methods of intervention (e.g., campaigns that target those attitudes).

Our response: Thank you for your comment. Although we agree that this would be interesting, we believe this is outside of the scope of the current manuscript and is the focus of a different manuscript in preparation.

Comment #11: The section of the abstract regarding offers of “a product” is difficult to follow. I had to look to the tables to understand which product was being referred to and why there were ranges of ORs presented. Please present these findings in a clearer manner.

Our response: Thank you for your comment. We have revised the Abstract to present the odds of being offered e-cigarettes and noted that similar characteristics were associated with offers of cigarettes.

Comment #12: Sensation seeking is typically assessed as a scale, rather than a single item. Is there precedent for using a single item? Has that approach between validated? Also, “ease of interpretation” is not an appropriate reason to recode responses.

Our response: Thank you for your comment. Unfortunately, only a single item was included in the questionnaire; as a result we were limited in using this single item in the analyses. A similar single-item measure has been used in the National Longitudinal Study of Adolescent Health (Add Health).

Comment #13: When proposing future directions for research, I suggest considering research that would better characterize who is doing the offering: friends? Family? Peers from their own social group or peers from other social groups? I believe that some of the articles on e-cigarettes discuss this issue. I’m not sure if the literature on offers of cigarettes discuss this issue.

Our response: Thank you for your suggestion. We have noted in the revised manuscript that future research should identify who is offering the product.

Round 2

Reviewer 3 Report

After reading the revision, I found that authors had addressed the major concerns, and the manuscript has been significantly improved.